# Knowledge, Attitude, Practices and Viewpoints of Undergraduate University Students towards Self-Medication: An Institution-Based Study in Riyadh

**DOI:** 10.3390/ijerph18168545

**Published:** 2021-08-12

**Authors:** Basheerahmed Abdulaziz Mannasaheb, Mohammed Jaber Al-Yamani, Sarah Abdulrahman Alajlan, Lamyaa Munahi Alqahtani, Shrouq Eid Alsuhimi, Razan Ibrahim Almuzaini, Abeer Fahad Albaqawi, Zahaa Majed Alshareef

**Affiliations:** 1Department of Pharmacy Practice, College of Pharmacy, AlMaarefa University, Riyadh 13713, Saudi Arabia; msuhail@mcst.edu.sa; 2College of Pharmacy, AlMaarefa University, Riyadh 13713, Saudi Arabia; 161220239@student.mcst.edu.sa (S.A.A.); 161220034@student.mcst.edu.sa (L.M.A.); 161220057@student.mcst.edu.sa (S.E.A.); 141220141@student.mcst.edu.sa (R.I.A.); 162220284@student.mcst.edu.sa (A.F.A.); 161220015@student.mcst.edu.sa (Z.M.A.)

**Keywords:** self-medication, AlMaarefa University, over-the-counter drugs, students, GPA, BMI

## Abstract

Rational and responsible self-medication (SM) is not only the key to better health outcomes, but also key to limiting adverse drug events. This institution-based cross-sectional study utilized seven- and four-item scales to assess the knowledge and attitude towards SM. Similarly, SM practices were measured using eight scale questions consisting of SM practice during the last six months, type of drug consumed, reason and frequency of SM, and so on. Statistical analyses were performed using SPSS. Overall, 371 students completed the questionnaire. The students with a good level of knowledge and positive attitude towards SM were 60.64% and 66.8%, respectively. About 55.5% of students practiced SM during the last six months using antipyretics (37.7%), multivitamins (36.4%), sleeping aids (20.2%), and anti-histamines (18.6%). Headache (79.2%), fever (37.7%), pain (25.9%), and colds and coughs (25.3%) were the illnesses for which they sought SM. The students admitted that drug side effects (75%), drug resistance (33.7%), drug interaction (41.5%), and poor treatment outcome (28.3%) were the consequences of irrational SM practice. Students (87.6%) propose that extending SM awareness through the Ministry of Health (83%) and pharmaceutical companies (48%) as major platforms would improvise the rational practice. Since AlMaarefa University students will be future healthcare professionals, their perception must be considered and accordingly educated to practice rational SM.

## 1. Introduction

We practice self-medication (SM) every day as a part of self-care of our health [1]. SM is an action practiced either personally or assisted by someone to manage unknown and less severe health conditions. The World Health Organization stated that the responsible practice of SM does pose the advantage of preventing and treating various mild health conditions that do not need initial medical consultation. Additionally, SM provides an affordable alternative for managing common illnesses [2]. However, inappropriate SM can lead to health consequences such as lack of drug effectiveness, allergy to certain medications, hindrance of actual diagnosis, increased resistance to certain types of medication, severe medication side effects, interaction with other drugs and food supplements, drug toxicity, overdose and dependency, development of resistance to certain drugs, withdrawal symptoms, and countless other health problems [3].

Although the objective of SM is clear, it is often practiced without proper scientific knowledge and background. In fact, SM practice could be forecasted in countries where healthcare expenses are met through people’s own pockets. Nevertheless, in Saudi Arabia, where healthcare expenses are supported by government and insurance companies, SM practice still prevails. Usually, the medications received from family or friends, leftover drugs from a previous prescription, and over-the-counter (OTC) drugs are self-administered to treat acute health conditions such as influenza, headache, coughs, and gastrointestinal disturbances [4,5]. SM is very common among the educated population, and the possible reasons for this rising tendency are the desire for self-care, kindheartedness towards family members in sickness [6], time trouble, shortage of health services, financial limitations, ignorance, misconceptions [7], and extensive advertisement and availability of drugs in non-drug shops [8]. Inappropriate SM not only harms the patient in the form of medication-related problems, but also increases treatment cost and frequency of hospital admission. SM is generally prevalent in all age groups; however, its extent differs among individuals and regions. The students of the healthcare program are expected to be more knowledgeable regarding the rational use of medications as compared to the general public. Rational use of medicine and the outcomes of irrational use are highlighted to these students in their curriculum [9]. SM among healthcare students would be because of their future medical preferences and knowledge about the drugs. Practicing SM, however, though seen by a lot of students as a time-managing, easy and successful process, does pose many hazards [10].

Various national [9,10] (Mustafa and Rohra [11]; Sultan MA et al. [12]; Mustafa SS et al. [13]) and international (Marwa AF et al. [14]; Lukovic et al. [15]; Helal RM and Abou-El Wafa HS [16]; Regina FA et al. [17]) studies have focused on SM behavior among college and university students, recognizing surge and widespread practice in this specific category, identifying key factors such as socio-demographic settings, lifestyle, easy accessibility and availability of medications, good knowledge, advertisement, and high level of education [18].

Moreover, the SM market is now on the rise [19]. During 2020, the OTC drugs market size was valued at more than $151 billion and anticipated to grow at a compound annual growth rate (CAGR) of over 5.1% between 2021 and 2027 ($209 billion). The COVID-19 outbreak has significantly affected the sales of OTC drugs with an increased focus on personal health during the pandemic. This has considerably amplified the use of cold and flu products, along with vitamins. The OTC products not only reduce treatment costs significantly, but also increase the affordability of treatment for all patient classes [20]. Various other scientific research studies have been conducted among AlMaarefa University (UM) students, such as the prevalence of stress [21], consumption of caffeine [22], sleep and academic performance [23], and COVID-19 related anxiety [24].

## 2. Materials and Methods

### 2.1. Study Design and Population

This institution-based cross-sectional study was conducted to investigate the level of knowledge, attitude, and practice of SM among undergraduate students of UM, Riyadh, Kingdom of Saudi Arabia. This study was approved by the institutional review board of UM with the registration number (03-02032021) and conducted over a period of four months from February to May 2021. Undergraduate students from the College of Pharmacy, the College of Medicine, and the College of Applied Sciences of UM were included. Students of other universities and those who disagreed to participate were excluded from this study.

### 2.2. Study Tool

The study employed a pre-designed validated structured questionnaire to collect the data, which was developed after an extensive literature review. The questionnaire was categorized into segments such as demography, knowledge, attitude, practice and student’s viewpoint and opinion regarding SM. The bilingual questionnaire was made available online to the students of UM through the university message center and other platforms. A consent form was displayed at the beginning of the questionnaire explaining the purpose of the study and assuring their identity and confidentiality. The participants had an option either to accept or reject participation.

### 2.3. Validity and Reliability of Study Tool

The content validation of the questionnaire was done by expert professors in the field. The face validity was tested by conducting two pilot studies on 30 students with a gap of one week (10 students each from the colleges of Pharmacy, Medicine, and Applied Sciences). These participants were excluded from the final sample. Necessary modifications were made based on the pre-test feedback. Additionally, Cronbach’s alpha factor (0.75) was calculated to check the questionnaire reliability. Repeat responses were identified by tracking student university ID and were excluded from data analysis.

### 2.4. Sampling Method and Sample Size

The sample size was determined using the single population proportion formula by assuming a 95% confidence level, 5% margin of error, and precision level of 5%. The proportion of SM observed in the pilot study was 66%. The number of students registered at UM was 2327. The correction formula was applied to calculate the final sample (300) size [25]. However, to ensure more representative data and anticipating a few incomplete and repeat responses, we collected a larger sample size of 399.
(1)$$\mathrm{Sample}\;\mathrm{size}=\frac{Z_{1-\propto 2}{}^{2}p\left(1-p\right)}{d^{2}}$$

### 2.5. Data Collection

Participant’s knowledge about responsible SM was assessed using 7 questions with a 7-point scale. All 7 questions were given a score of 1 or 0 (each correct response had a value of 1 and incorrect or do not know was given a value of 0). Assessment of participant’s knowledge about SM consisted of responsibility towards SM, when to discontinue SM drugs, antibiotics for self-use, and class of drugs for self-use. The expected maximum cumulative score was 7 and the minimum was 0. Participant’s knowledge was categorized as good, moderate, and poor using the original Bloom’s cut-off points (Good, 80–100%; Moderate, 50–79%; and Poor, <50%). A score of 6–7 points was considered good, moderate if the score was 4–5, and poor for a score of ≤3. Likewise, attitude towards responsible SM was assessed by putting 4 statements on a Likert’s scale, from strongly agree (5), agree (4), neutral (3), disagree (2), to strongly disagree (1). The maximum expected score was 20 and the minimum was 4. The level of attitude was classified using modified Bloom’s cut off points, as positive if the score was between 16–20 (80–100%), neutral for a score of 12–16 (60–79%), and negative if a score was less than 12 (<60%).

Students’ practice towards SM was measured using 8 questions that consisted of SM practice during the last six months, type of drug consumed, the reason for and frequency of SM, source of drug information, place of obtaining the drug, and negative impact of irrational SM. Students shared their viewpoints and opinions to rationalize and improvise the responsible SM and to spread the awareness towards SM through various platforms.

### 2.6. Data Processing and Analysis

The online responses collected were subjected to tests for completeness and consistency before processing for analysis. Incomplete and repeat responses were excluded from the analysis. The descriptive statistics were summarized by measuring frequency, percentage, and standard deviation. Variation in the adequacy of knowledge and attitude was hypothesized among students of various colleges; cross-tabulation was performed applying the chi-square test. Multiple stepwise linear regressions were applied to find out the relationship between scores of knowledge and attitude with participant’s characteristics. The analyzed data was systematically organized and presented in tabular, graphical, and narrative forms. SPSS (version 27.0, IBM, New York, NY, USA) was used to perform all statistical analyses. A *p*-value of less than 0.05 was considered significant.

## 3. Results

### 3.1. Demographic Information

In total, 399 students responded to the survey. After excluding the incomplete and repeat responses, a total of 371 responses were included for data analysis. Students from the College of Pharmacy were the highest participants in this survey, accounting for 153 (41.2%), followed by Applied Sciences with 130 (35%), and the College of Medicine students accounting for 88 (23.7%). Female respondents were a major contributor (61.5%), and the major age group was 20–25 years (70.6%). Concerning nationality, 77.1% were Saudis and 87.3% of respondents were unmarried (Table 1).

### 3.2. Student’s Level of Knowledge towards SM

Altogether, the mean knowledge score of participants was 4.71 (SD = 1.5). The respondents with a good, moderate, and poor level of knowledge towards SM were 225 (60.64%), 115 (31%), and 31 (8.35%), respectively. Factors significantly impacting the level of SM knowledge were students’ College, age, study level, BMI, nationality, student situation, and family member working in the health sector. The majority of participants with good knowledge were from the College of Pharmacy (18.9%). The mean knowledge score of students with normal BMI was significantly higher (4.93) compared to overweight and underweight students. Students belonging to study levels 7 and 8 showed a significantly (*p* < 0.001) higher level of knowledge score (5.5) compared to students in other levels of study. The average knowledge score of students whose family members work in the healthcare sector was significantly (*p* = 0.02) higher (4.94) compared to students who do not have any family members in the health sector. Similarly, divorced or separated participants had non-significantly higher mean knowledge scores (5.33) compared to single and married participants. No significant impact of marital status, gender, and the presence of chronic disease was observed on the knowledge score (Table 1). The mean knowledge score of students from the colleges of Pharmacy, Medicine, and Applied Sciences were 5.18, 4.81, and 4.1, respectively (Table 2).

### 3.3. Student’s Attitude towards SM

Overall, the mean attitude score of participants was 16.4 (SD = 2.06). The number of participants with a positive, neutral, and negative attitude towards SM was 248 (66.84%), 108 (29.11%), and 15 (4.04%), respectively. Interestingly, numbers of participants (66.84%) with a positive attitude were higher compared to students with good knowledge (60.64%). The average attitude score of female students was significantly (*p* < 0.019) higher (16.6) compared to males. The students staying in university accommodation have shown a significantly (*p* < 0.001) higher positive (17.7) attitude towards SM compared to students who stay with their family or outside of university accommodation. Students of study levels 9 and 10 have shown significantly (*p* < 0.006) higher levels of attitude scores (17.3) compared to students in other levels of study. Likewise, the average attitude score of full-time students was significantly (*p* < 0.006) higher (16.5) compared to working students. However, students with lower GPA (1–2), have shown a non-significantly higher (17.45) score of attitudes compared to students with average and higher GPA (Table 1). Similarly, the overall average attitude scores of students from the College of Pharmacy (16.7), aged more than 25 years (16.7), of Saudi nationality (16.5), and suffering from chronic disease (16.9) were non-significantly higher compared to their counterparts (Table 2).

### 3.4. Self-Medication (SM) Practice

During the last six months, 55.5% of students have practiced SM. About half of the participants (44.5%) practiced SM once during the last six months; whereas, 34% of students mentioned they used SM during the previous month.

### 3.5. Information and Place of Obtaining the Drugs for SM

Table 3 depicts the sources utilized by the students to get the information about medications before practicing SM. Almost half of the students (53.9%) reported academic knowledge and books as a primary source of drug information, followed by previous prescriptions (40.4%) and the internet (29.6%). Almost all respondents (93.5%) mentioned pharmacy stores as a prime place for obtaining medications for self-care.

### 3.6. Medications Used for SM

The most common class of drugs used during the last six months were antipyretics (37.7%), multivitamins (36.4%), sleeping aids (20.2%), anti-histamines (18.6%), and anti-inflammatories (17.3%); however, use of Central Nervous System (CNS) stimulants (1.1%) was least common (Figure 1).

The participants mentioned that headache (79.2%), fever (37.7%), pain (25.9%), colds and coughs (25.3%), and menorrhea (23.2%) were the illnesses for which they sought SM (Figure 2).

### 3.7. Reasons for Opting SM Practice

Figure 3 represents the major reasons mentioned by students that influenced them to practice SM. About 62% of students stated low severity of illness as the main reason to practice SM. However, 33.2% of students declare that they use SM as an active role in managing their health problems. About 23.7% of participants admit that they were too busy to visit the healthcare facility, hence opted for SM. Other reasons such as long waiting time (7.5%) and learning opportunities (4.3%) were a few other reasons.

### 3.8. Negative Impact of Irrational SM

Figure 4 describes the students’ viewpoint on the negative impact of irrational SM. The participants admitted that drug side effects (75%), return of symptoms (22.6%), drug resistance (33.7%), drug interaction (41.5%), and poor treatment outcome (28.3%) are major consequences of irrational SM practice. Unfortunately, few participants (8.9%) mention that irrational SM does not have any negative impact.

### 3.9. Awareness towards Responsible SM

The study participants have shared their viewpoints and opinions to rationalize and improve responsible SM and to increase the awareness regarding SM. Interestingly, 87.6% of students proposed that extending awareness and education regarding the implications of SM would improve the rational SM practice. Similarly, enforcing strict rules regarding misleading pharmaceutical advertising (34.5%), working towards making health care facilities easily available (32.3%), and drug take-back centers for leftover or old drugs (19.1%) would have a significant impact on responsible SM, according to the participants of this study. The students have also mentioned the role of a key platform, such as the Ministry of Health (83%), pharmaceutical companies (48%), social media (44.7%), government (44.4%), and university (34.8%) (Figure 4), in spreading awareness regarding responsible SM.

### 3.10. Correlation between Knowledge and Attitude

The students with a negative attitude (<12 scores) had an average knowledge score of 4.23, whereas students with a positive attitude (16–20 score) had a 4.96 score, showing no significant change in knowledge score with a change of attitude. However, a positive correlation was noted between knowledge and attitude. There was a significant increase in attitude score with an increase in knowledge score (correlation coefficient (*r*) = 0.231 **, *p* = 0.000). Further analysis was conducted to validate these findings. As predicted, there was an association between good knowledge and a positive attitude towards SM (OR: 1.46; 95% Cl: 0.913–2.35).

### 3.11. Factors Influencing Self-Medication Practice

Identification of predictors influencing SM practice was determined using binary logistic analysis. Gender, nationality, and student’s situation did not show any significant association with SM practice. On the other hand, we observed a significant relationship between family members working in the healthcare sector and SM practice (*p* = 0.011). The students whose family members work in the health sector were 0.5 times more likely to not indulge in SM (OR: 0.583) compared to students with no family members working in the health sector. There was a significant relationship between the respondent suffering from long-term disease and the practice of SM (*p* = 0.005). Students who do not suffer from any long-term disease were 2.61 times more likely to not practice SM compared to students who suffer from chronic disease (OR: 2.610). Interestingly, we also observed a significant impact of college on SM practice (*p* = 0.033). The College of Pharmacy students has a 0.633 times higher chance of not indulging in the practice of SM compared to Medicine and Applied Sciences colleges. This could be due to better knowledge about medications (Table 4).

### 3.12. Factors Affecting Student’s Knowledge and Attitude Regarding SM

The total score of knowledge and attitude (KA) is a dependent variable. The multiple stepwise linear regressions were conducted using a single statistically significant factor derived from univariate analysis of influencing factors which are as follows: gender, college, age, study level, GPA, BMI, residential status, nationality, marital status, students’ situation, family member working in the health sector, and suffering from any chronic disease. It was evident from the results of multiple linear regression analysis that students’ age, college, gender, level of study, nationality, and presence of the chronic disease influenced the total scores of knowledge and attitude (*p* < 0.05). The students’ nationality and college of study have a negative significant impact on total knowledge and attitude score, whereas all other predictors were contributing positively to the total KA score (Table 5).

## 4. Discussion

Rational and responsible SM is key not only for better health outcome, but also to limit adverse drug events, dosage, treatment errors, and risk of addiction or abuse. This study was conducted in students of UM, considering that health science students have adequate knowledge about medicines; hence, they are more aware of the consequences of improper SM. The UM student’s perception about SM will be valuable in judging the characteristics of their future medication handling, promoting health, and reducing drug-related problems.

In the present study, students’ overall adequacy of knowledge and attitude towards SM were satisfactory. Although SM has many pros and cons, it depends on who uses it and how it is used for self-treatment [26]. In our study, we noticed that 55% of UM students took SM drugs during the last six months, which is lower compared to studies of students of King Khalid University, Abha [12] (98.7%), Al-Qassim University [13] (86.6%), Jazan University [10] (87%), Bangladeshi Undergraduate Pharmacy Students [27] (88%), Zabol University of Medical Sciences, Iran [28] (57%), Kasturba Medical College, India [29] (78.6%) and higher compared to students of Imam Abdulrahman Bin Faisal University in Dammam [9] (26%). Comparing the results with findings from Serbia [15], where the prevalence of SM was dependent on age and gender, our study showed no significant difference. We observed the significant impact of college (Pharmacy), the presence of long-term disease, and family members working in health on low liability to indulge in SM practice. The common reason for the high prevalence of SM might be academic knowledge about drugs.

The study shows that 292 (78.7%) of the students are aware of responsible SM, similar to a previous study done in Kasturba Medical College, India [29]. Like other previously published studies [10,13,16,27,28,29], headache, fever, pain, cold, cough, and menorrhea were the most common illnesses for which SM was practiced. The most common type of medication preferred for self-use was antipyretic (37.7%), multivitamins (36.4%), antihistamines (18.6%), and anti-inflammatories (17.3%). Similar findings were reported in a study conducted on students of a Portuguese University [17], and studies within Saudi Arabia [9,12]. The reasons mentioned by our students for opting for medication for self-care were minor illness (62%), previous experience (33.2%), and lack of time to attend the health care facility (23.7%). These findings are comparable with the outcomes reported by earlier studies [9,12,16]. Improper SM practices may lead to the advancement of the disease, incorrect diagnosis, and promote serious health hazards. Easy availability of medicine, quick relief, and time-saving were the contributing factors for SM as reported by other studies [29,30]. Contrary to antipyretics and antihistamines, the use of antibiotics was much lower (12.4%). This practice indicates that students of UM have adequate knowledge about antimicrobial resistance and the consequences of irrational use of antibiotics. A similar finding has been quoted in a previous study [27]. Our study has noticed that the pharmacy store was commonplace for obtaining a drug for SM. The key factor for SM practice was student’s adequate medication knowledge obtained during their course of study. These findings are similar to those reported in earlier studies [12,27,28]. However, a study conducted on students of the city of Mansoura, Egypt [16] quoted academic knowledge was the least important contributing factor. Our study noticed that the majority of respondents practiced SM once a month, which is comparable with the study conducted on students at Dammam University [9]. The respondents of the present study have mentioned drug side effects, interaction, and resistance as major negative outcomes of irrational and unsupervised SM, which is contrary to the findings reported in the Zabol University of Medical Sciences, Iran [28], where the highest number of respondents stated no negative impact of SM. Interestingly, students have shared their viewpoints and opinions regarding the rationalization and improvement of responsible SM and the conveying of awareness. A majority of respondents agree that creating awareness and education regarding the implication of self-medication would improvise and rationalize SM practice. This finding was comparable with the reports mentioned by students of Kasturba Medical College, India [29]. The students have also mentioned the role of key platform such as the Ministry of Health and pharmaceutical companies in spreading the awareness regarding responsible SM.

The adequacy of knowledge among students of UM is satisfactory, unlike findings of King Khalid University [12], where student’s knowledge about SM was poor. The student’s age, gender, study level, college of study, BMI, full/working student, and family member in the healthcare field had a significant impact on the level of knowledge. The level of knowledge increases dramatically as students progress to the higher study levels of the program. The study also showed that College of Pharmacy students (18.9%) had better knowledge regarding SM compared to students of the College of Medicine and Applied Sciences. Likewise, the respondent’s attitude towards responsible SM was greatly positive. In the present study, we observed the significant impact of gender, study level, full/working student, residential status, and marital status on attitude. As students progress to higher study levels of the program, a greater positive attitude towards SM has been observed. These observations are in line with studies conducted in other parts of Saudi Arabia [9,12]. In contrast, a study conducted on students of Taibah University [31] observed a high level of negative attitudes towards SM. We observed a positive correlation between knowledge and attitude scores. An important finding in our study is that students of UM refer to books and academic knowledge to get medication information and retain a higher level of consciousness regarding the negative impact of irrational SM. Health professionals should play a predominant role in guiding self-medication behaviors in the general public.

### 4.1. Strengths and Limitations

A high response rate (399) was one of the strengths of this study. Students from all the study levels were accepted. Since the practice segment of the questionnaire was embedded with recall questions on drug use, the time period was restricted to the last six months to limit the recall bias. Considering the complexity of some survey questions, an online survey was set up bilingually (English and Arabic) to improve the understanding of questions. Since our study is cross-sectional, it is recommended to carry out prospective studies on student behavior regarding SM and determines factors influencing attitude and practice. Many respondents were busy with their exams, labs, and lectures during our study period and physical interaction was limited due to COVID-19 precautionary measures, hence data collection was strenuous. Although more than the required number of respondents completed the survey, many students still did not participate in our study.

### 4.2. Recommendations

Although students of UM reported having satisfactory and adequate perception regarding SM, it is suggested that offering an elective course highlighting the responsibility towards rational SM and the negative impact of irresponsible SM would be useful. Additionally, strategies to conduct small group work or workshops culminating in the impact of SM could be developed. We recommend the expansion of this study and encourage all the students of UM to complete the survey.

## 5. Conclusions

The overall knowledge and attitude scores of UM students were satisfactory; two-thirds of participants had good knowledge and a better attitude towards SM. Moreover, SM was practiced by half of the respondents during the last six months, which indicates their awareness of responsible SM. It is suggested that this study be expanded to the remaining students of UM to support these findings. All the students of health care programs and especially Pharmacy students have an important role to play in the SM topic and its rational practice. Since students of UM will be future healthcare professionals and are going to contribute immensely to the public healthcare system, their perception has to be considered and accordingly educated to practice rational and responsible SM.

## Figures and Tables

**Figure 1 ijerph-18-08545-f001:**
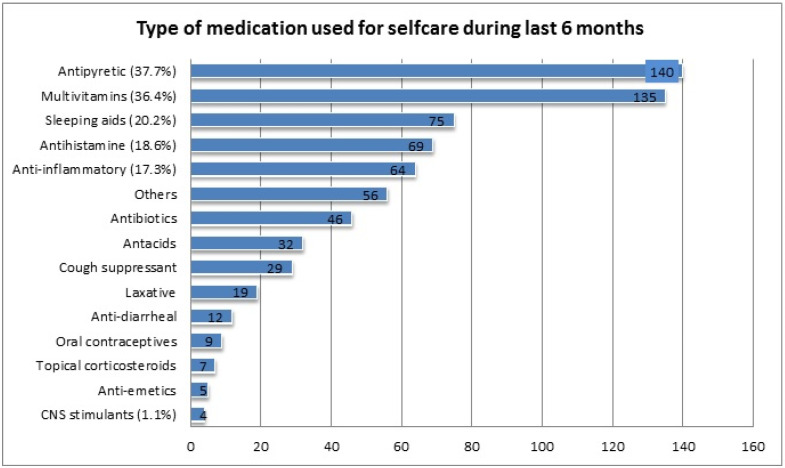
Type of medication used for self-care during the last six months.

**Figure 2 ijerph-18-08545-f002:**
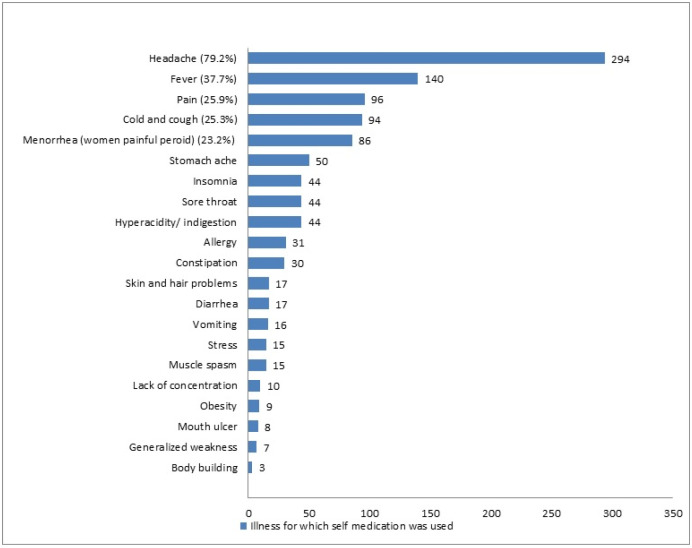
Illness for which SM was practiced during the last months.

**Figure 3 ijerph-18-08545-f003:**
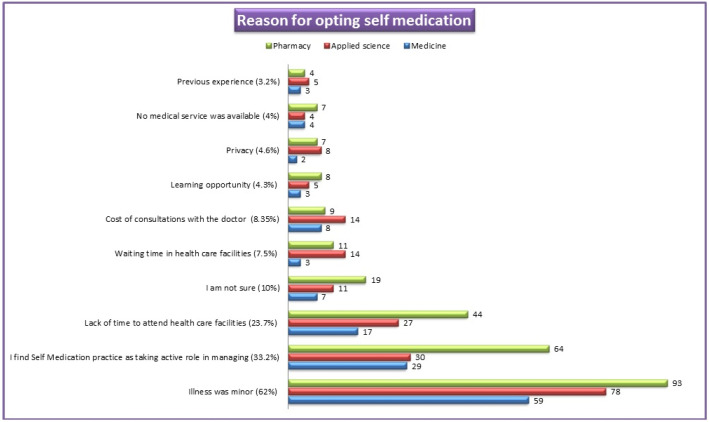
Reasons for opting self-medication practice.

**Figure 4 ijerph-18-08545-f004:**
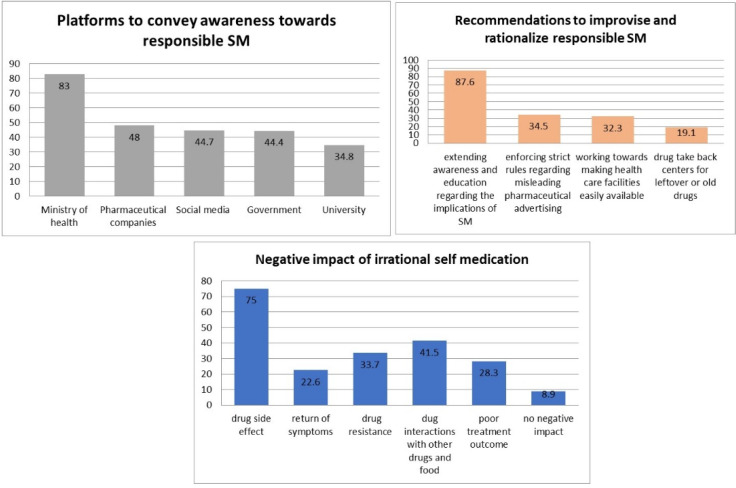
The negative impact of irrational SM, awareness, recommendations to improvise and rationalize responsible SM.

**Table 1 ijerph-18-08545-t001:** Socio-demographic characteristics and their association with the adequacy of knowledge and attitude towards self-medication (SM).

Demographic Variable		Demographic Characteristics (%)	Adequacy of Knowledge (*n* = 371)				Adequacy of Attitude (*n* = 371)			
			Good (%)	Moderate (%)	Poor (%)	*p* Value	Positive (%)	Neutral (%)	Negative (%)	*p* Value
Colleges	Pharmacy	153 (41.2)	70 (18.9)	68 (18.3)	15 (4)	0.001 *	81 (21.8)	69 (18.6)	3 (0.8)	0.639
	Medicine	88 (23.7)	27 (7.3)	48 (12.9)	13 (3.5)		37 (10)	46 (12.4)	5 (1.3)	
	Applied Sciences	130 (35.1)	26 (7)	57 (15.4)	47 (12.7)		60 (16.2)	63 (17)	7 (1.9)	
Age	<20 years	30 (8.1)	2 (0.5)	16 (4.3)	12 (3.2)	0.001 *	10 (2.7)	18 (4.9)	2 (0.5)	0.083
	20–25 years	262 (70.6)	84 (22.6)	129 (34.8)	49 (13.2)		122 (32.9)	134 (36.1)	6 (1.6)	
	>25 years	79 (21.3)	37 (10)	28 (7.5)	14 (3.8)		46 (12.4)	26 (7)	7 (1.9)	
Nationality	Saudi	286 (77.1)	110 (29.6)	122 (32.9)	54 (14.6)	0.017 *	144 (38.8)	129 (34.8)	13 (3.5)	0.130
	Non-Saudi	85 (22.9)	13 (3.5)	51 (13.7)	21 (5.7)		34 (9.2)	49 (13.2)	2 (0.5)	
Student situation	Full time student	291 (78.4)	88 (23.7)	146 (39.4)	57 (15.4)	0.006 *	146 (39.4)	138 (37.2)	7 (1.9)	0.006 *
	Working student	80 (21.6)	35 (9.4)	27 (7.3)	18 (4.9)		32 (8.6)	40 (10.8)	8 (2.2)	
Family member in health sector	Yes	218 (58.8)	83 (22.4)	103 (27.8)	32 (8.6)	0.02 *	108 (29.1)	99 (26.7)	11 (3)	0.896
	No	153 (41.2)	40 (10.8)	70 (18.9)	43 (11.6)		70 (18.9)	79 (21.3)	4 (1.1)	
Gender	Male	143 (38.5)	53 (14.3)	53 (14.3)	37 (10)	0.051	57 (15.4)	77 (20.8)	9 (2.4)	0.019 *
	Female	228 (61.5)	70 (18.9)	120 (32.3)	38 (10.2)		121 (32.6)	101 (27.2)	6 (1.6)	
Study level/phase	Level 1–2	42 (11.3)	9 (2.4)	21 (5.7)	12 (3.2)	0.001 *	18 (4.9)	23 (6.2)	1 (0.3)	0.006 *
	Level 3–4	102 (27.5)	16 (4.3)	48 (12.9)	38 (10.2)		48 (12.9)	43 (11.6)	11 (3)	
	Level 5–6	83 (22.4)	26 (7)	46 (12.4)	11 (3)		43 (11.6)	39 (10.5)	1 (0.3)	
	Level 7–8	71 (19.1)	39 (10.5)	26 (7)	6 (1.6)		24 (6.5)	47 (12.7)	zero	
	Level 9–10	43 (11.6)	23 (6.2)	17 (4.6)	3 (0.8)		31 (8.4)	12 (3.2)	zero	
	Level 11–12	20 (5.4)	9 (2.4)	9 (2.4)	2 (0.5)		10 (2.7)	9 (2.4)	1 (0.3)	
	Internship students	10 (2.7)	1 (0.3)	6 (1.6)	3 (0.8)		4 (1.1)	5 (1.3)	1 (0.3)	
GPA	1–2	31 (8.4)	8 (2.2)	15 (4)	8 (2.2)	0.509	20 (5.4)	11 (3)	zero	0.291
	2–3	154 (41.5)	63 (17)	62 (16.7)	29 (7.8)		67 (18.1)	82 (22.1)	5 (1.3)	
	3–4	186 (50.1)	52 (14)	96 (25.9)	38 (10.2)		91 (24.5)	85 (22.9)	10 (2.7)	
BMI	Below 18.5	35 (9.43)	7 (1.9)	17 (4.6)	11 (3)	0.023 *	26 (7)	9 (2.42)	zero	0.388
	18.5 to 24.9	200 (53.9)	77 (20.8)	92 (24.8)	31 (8.4)		132 (35.6)	62 (16.7)	6 (1.6)	
	25 to 29.9	93 (25.1)	20 (5.4)	47 (12.7)	26 (7)		59 (15.9)	27 (7.3)	7 (1.9)	
	30 and above	43 (11.6)	19 (5.1)	17 (4.6)	7 (1.9)		31 (8.4)	10 (2.7)	2 (0.53)	
Residential status	Living with family	299 (80.6)	93 (25.1)	141 (38)	65 (17.5)	0.424	137 (36.9)	155 (41.8)	7 (1.9)	0.001 *
	University accommodation	10 (2.7)	3 (0.8)	7 (1.9)	zero		10 (2.7)	zero	zero	
	Non-university accommodation	62 (16.7)	27 (7.3)	25 (6.7)	10 (2.7)		31 (8.4)	23 (6.2)	8 (2.2)	
Marital status	Single	324 (87.3)	102 (27.5)	155 (41.8)	67 (18.1)	0.775	159 (42.9)	155 (41.8)	10 (2.7)	0.009 *
	Married	41 (11.1)	18 (4.9)	15 (4)	8 (2.2)		18 (4.9)	18 (4.9)	5 (1.3)	
	Divorced/separated	6 (1.6)	3 (0.8)	3 (0.8)	zero		1 (0.3)	5 (1.3)	zero	
Presence of long term/chronic disease	No	324 (87.3)	95 (25.6)	159 (42.9)	70 (18.9)	0.804	151 (40.7)	160 (43.1)	13 (3.5)	0.053
	Yes	47 (12.7)	28 (7.5)	14 (3.8)	5 (1.3)		27 (7.3)	18 (4.9)	2 (0.5)	

Pearson chi-Square; * *p* value < 0.05. Abbreviations: GPA: Grade Percentage Average; BMI: Body Mass Index.

**Table 2 ijerph-18-08545-t002:** Mean knowledge and attitude scores against demographic variables.

**Demographic Variable**		**Mean Knowledge Score (7-Point Scale)**	**Mean Attitude Score (20-Point Scale)**
Colleges	Pharmacy	5.18	16.72
	Medicine	4.81	16.04
	Applied Sciences	4.1	16.3
Age	<20 years	3.7	15.96
	20–25 years	4.69	16.37
	>25 years	5.17	16.73
Nationality	Saudi	4.83	16.51
	Non-Saudi	4.3	16.1
Student situation	Full time student	4.65	16.51
	Working student	4.93	16.06
Family member in health sector	Yes	4.94	16.41
	No	4.39	16.41
Gender	Male	4.66	16.1
	Female	4.75	16.6
Study level/phase	Level 1–2	4.21	16.11
	Level 3–4	3.99	16.02
	Level 5–6	4.85	16.66
	Level 7–8	5.5	16.29
	Level 9–10	5.27	17.3
	Level 11–12	5.1	16.5
	Internship students	4.3	16.5
GPA	1–2	4.45	17.45
	2–3	4.85	16.25
	3–4	4.64	16.37
BMI	Below 18.5	4.24	16.48
	18.5 to 24.9	4.93	16.47
	25 to 29.9	4.4	16.26
	30 and above	4.73	16.46
Residential status	Living with family	4.63	16.4
	University accommodation	5	17.7
	Non-university accommodation	5.06	16.27
Marital status	Single	4.66	16.42
	Married	5.02	16.36
	Divorced/separated	5.33	16.5
Presence of long term/chronic disease	No	4.6	16.34
	Yes	5.51	16.89

**Table 3 ijerph-18-08545-t003:** Self-medication (SM) practice among students of UM.

Self-Medication Practice Information	Sample (*N*)	Percentage
During the last six months did you self-medicate yourself?		
Yes	206	55.5
No	165	44.5
Frequency of practice (I used self-medication drug)		
During last month	126	34.0
During the last three months	80	21.6
During the last six months	165	44.5
Place of obtaining the medication for self-use		
Pharmacy store	347	93.53
Family/friends	60	16.17
Supermarket	20	5.39
Internet/online	27	7.27
Source of drug information		
Previous prescription	150	40.43
Friends/relatives	50	13.47
Media	25	6.73
Internet	110	29.64
Parent	30	8.08
Academic knowledge and book	200	53.9
Other	33	8.89

**Table 4 ijerph-18-08545-t004:** Factors influencing self-medication practice.

		Practice of Self-Medication During Last Six Months		*p*-Value	Cremer’s V	Odds Ratio (Risk Estimate)		
Category	Subgroup					95% confidence interval		
		No (%)	Yes (%)	0.011 *	0.132	Value	Lower	Upper
Family member working in health sector?								
	Yes	85 (22.9)	133 (35.8)			0.583	0.384	0.886
	No	80 (21.6)	73 (19.7)					
Do you suffer from any long term/chronic disease?	Yes	12 (3.2)	35 (9.4)	0.005 *	0.145	2.610	1.308	5.208
	No	153 (41.2)	171 (46.1)					
Colleges	Pharmacy	58 (15.63)	95 (25.60)	0.033 *	0.111	0.633	0.416	0.965
	Medicine & Applied Sciences	107 (28.8)	111 (29.9)					
Gender	Male	63 (17)	80 (21.6)	0.898	0.007	0.973	0.639	1.482
	Female	102 (27.5)	126 (34)					
Nationality	Saudi	130 (35)	156 (42)	0.486	0.036	1.190	0.729	1.945
	Non-Saudi	35 (9.4)	50 (13.5)					
Student situation	Full time student	127 (34.2)	164 (44)	0.539	0.032	0.856	0.521	1.406
	Working student	38 (10.2)	42 (11.3)					

Pearson chi-Square; * *p* value < 0.05.

**Table 5 ijerph-18-08545-t005:** Multiple stepwise linear regression analysis of factors affecting student’s knowledge and attitude towards SM.

Independent Variable	Regression Coefficient β	Standard Error	Standardized Regression Coefficient β	t	*p*-Value
Constant term	18.756	1.129	---	16.616	0.000
College	−0.687	0.164	−0.212	−4.189	0.000
Nationality	−0.851	0.336	−0.127	−2.534	0.012
Age	0.967	0.284	0.180	3.400	0.001
Gender	1.111	0.292	0.191	3.801	0.000
Level of study	0.254	0.096	0.135	2.641	0.009
Do you suffer from any long term/chronic disease?	0.123	0.051	0.115	2.394	0.017

## Data Availability

The datasets generated and/or analyzed during the current study are not publicly available due to ethical restrictions but are available from the corresponding author on reasonable request.

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
