# Peer review of "Knowledge, Attitude, Practices and Viewpoints of Undergraduate University Students towards Self-Medication: An Institution-Based Study in Riyadh"

_ijerph, 2021, doi:10.3390/ijerph18168545_

Round 1
Reviewer 1 Report
The article "Knowledge, Attitude, Practices and viewpoints of undergraduate university students towards self-medication: An institution based study in Riyahd" provides a good contribution to international research.
The introduction explains adequately enough the reason behind the development of this study.
The methodology is appropriate for the proposed research design.
The discussion and conclusions highlight the main points to focus on, as well as the main limitations related to this research.
I suggest modifying the introduction, to make it more complete and to increase the curiosity of readers.
Author Response
Reviewer 1:
Reviewer’s comment: I suggest modifying the introduction, to make it more complete and to increase the curiosity of readers.
Response 1: Introduction part has been modified to make it more complete.
Reviewer 2 Report
This is a study on self-medication practices and knowledge of these in health science students.
These are two surveys on the knowledge of the atom-medication and the knowledge of the same, it is a methodologically adequate study with reasonable results and that has the limitations of a cross-sectional study.
Author Response
Reviewer’s comment: This is a study on self-medication practices and knowledge of these in health science students. These are two surveys on the knowledge of the atom-medication and the knowledge of the same, it is a methodologically adequate study with reasonable results and that has the limitations of a cross-sectional study.
Response 1: Moderate English changes have been made wherever necessary and Introduction part has been modified to increase the reader’s curiosity.
Reviewer 3 Report
Thank you for the opportunity to review this paper. This reviewer believes assessing the Knowledge, Attitude, Practices, and viewpoints of students towards self-medication in Riyadh is interesting and publication-worthy. The manuscript is clear and easy to understand. The researchers applied an online survey among AlMaarefa University (UM) students. 399 out of 2327 UM students participate in the survey. The reviewer believes this manuscript needs a few major edits and minor edits.
Major comments:
Please discuss potential selection bias in this survey. There are 399 out of 2327 students choosing to answer the survey. Is there any difference between the study cohort and the student body of UM?
Consider reducing the total number of figures and tables.
For example, combine information presented in Table 1,2,3, and Figure 1 into one summary table. The labels of Figure 2 are not readable. Please consider presenting it using a vertical figure or present it with a table.
Similar comments for table 4 and figure 2.
For example, combine figure 6,7,8 into one figure (three panels)
Minor:
Make the color of tables consistent
OR of “family member working in health sector” in table 6 is placed in the wrong line. Please correct it.
Author Response
Major comments:
Reviewer’s comment1: Please discuss potential selection bias in this survey. There are 399 out of 2327 students choosing to answer the survey. Is there any difference between the study cohort and the student body of UM?
Response 1: All the students of different colleges of UM, were approached to participate in this study, as mentioned in methodology; survey was shared through online platform due to covid-19 restrictions and students attending their classes online. We assume low probability of selection bias as all the students of UM had equal opportunity to participate. Since this study was carried out with the involvement of college of pharmacy students, it was easy to reach out to college of pharmacy students through various platforms and encourage them to participate. Hence high numbers of participants are from college of pharmacy. Nevertheless other colleges too have participated satisfactorily.
Reviewers comment 2: Consider reducing the total number of figures and tables.
For example, combine information presented in Table 1,2,3, and Figure 1 into one summary table. The labels of Figure 2 are not readable. Please consider presenting it using a vertical figure or present it with a table. Similar comments for table 4 and figure 2. For example, combine figure 6,7,8 into one figure (three panels)
Response 2: Number of tables and figures have been compiled and reduced to maximum possible extent. Numbers of figures are reduced to 4 from 8. Numbers of tables are reduced to 6 from 7.
Minor comment
Reviewers comment 3: Make the color of tables consistent
OR of “family member working in health sector” in table 6 is placed in the wrong line. Please correct it.
Response 3: All the suggested changes have been done.
Round 2
Reviewer 3 Report
Figures and tables are significantly improved. Please consider to further reduce the number of tables.
Selection bias exists in this study, even everyone has access to participant this study. For example, students who didn't answer the survey may not have experience with self medication. Those students may be not interested in SM or have negative attitude to SM. This study may overestimate knowledge, attitude and practice of SM among UM students. Please further discuss in the manuscript.
Author Response
Dear reviewer
Thanks for your valuable feedback and comments
- Extensive editing of English language and style has been done with the help of native English speaker and online Grammarly program.
- Data of table 3 is merged with table 1, hence one more table is deducted.
- Results and conclusion part is revised.
Regarding selection bias, we understand your point of view. As mentioned in earlier reply, survey was open for all the students of UM, every student received this survey through official university message center via email. We mentioned in strength and limitation part that many students didn’t wish to participate due to their busy college schedule. It is also mentioned in the limitations and recommendation part to expand this study and to collect the data from almost all the students of UM.
As you mentioned the students who didn’t participate in our study, may be not interested in SM or have negative attitude to SM or it could be vise versa. we will be able to conclude it only after we get their responses.
Of course, it is not possible to collect the data from all the population of study, that is why we calculate sample size, and we achieved the desired sample.
Hope this answers your query